# Influence of Electronic Cigarette Characteristics on Susceptibility, Perceptions, and Abuse Liability Indices among Combustible Tobacco Cigarette Smokers and Non-Smokers

**DOI:** 10.3390/ijerph16101825

**Published:** 2019-05-23

**Authors:** Cosima Hoetger, Rose S. Bono, Nicole E. Nicksic, Andrew J. Barnes, Caroline O. Cobb

**Affiliations:** 1Department of Psychology, Virginia Commonwealth University, 806 W Franklin St, Richmond, VA 23284, USA; hoetgerc@vcu.edu; 2Center for the Study of Tobacco Products, Virginia Commonwealth University, 100 W Franklin St, Richmond, VA 23220, USA; 3Department of Health Behavior and Policy, Virginia Commonwealth University, 830 E Main St, Richmond, VA 23219, USA; rose.bono@vcuhealth.org (R.S.B.); Nicole.Nicksic@vcuhealth.org (N.E.N.); andrew.barnes@vcuhealth.org (A.J.B.)

**Keywords:** electronic cigarette, flavors, policy, purchase task, risk perceptions, tobacco regulation

## Abstract

This study assessed how electronic cigarette (ECIG) characteristics amenable to regulation—namely nicotine content, flavor, and modified risk messages—impact ECIG use susceptibility, harm/addiction perceptions, and abuse liability indices among combustible tobacco cigarette (CTC) smokers and non-smokers. CTC smokers and non-smokers varying in ECIG use recruited via Amazon Mechanical Turk (MTurk) completed an online survey in 2016 (analytic *n* = 706). Participants were randomly assigned to one of eight conditions differing in ECIG characteristics: nicotine content (no, low, high), flavor (menthol, tobacco, fruit), or modified risk message (reduced harm, reduced carcinogen exposure). Regressions assessed ECIG susceptibility, harm/addiction perceptions, and abuse liability indices (purchase task measures of breakpoint/intensity) within each regulatory domain (nicotine content, flavor, message) and their interactions with CTC/ECIG status. Differential effects on ECIG susceptibility, harm/addiction perceptions, and abuse liability indices were observed by regulatory domain with many effects moderated by CTC/ECIG status. ECIG nicotine content and flavor conditions were the most influential across outcomes. Greater nicotine content, tobacco-flavored and reduced carcinogen exposure ECIGs were more highly preferred by CTC smokers with some differing preferences for non-users. Findings reinforce consideration of discrete ECIG preferences across tobacco use status to improve regulatory efficacy.

## 1. Introduction

Recent increases in the use of alternative tobacco products, such as electronic nicotine delivery devices or electronic cigarettes (ECIGs), especially among youth and young adults in the United States (US) and elsewhere have raised concerns about the public health impact of tobacco use [1,2]. During 2016, 15.4% of US adults reported having used ECIGs at some point, with individuals aged 18–24 representing nearly one quarter of these users [3]. In 2014, almost half (47.6%) of combustible tobacco cigarette (CTC) smokers from a national US sample reported having used ECIGs at some point [4]. However, there is limited data on the short and long-term health effects of ECIGs as well as their utility for CTC cessation [5,6]. Some human and animal models have suggested that ECIG use may be associated with detrimental health outcomes [5] including, but not limited to, negative effects of some ECIG liquid solution flavorings on the pulmonary system [7,8], the presence of potentially toxic compounds within ECIG liquid solutions [9], as well as an increased risk of oxidative stress [10] and decreased immune function related to ECIG aerosol exposure [11]. Relatedly, some ECIGs are capable of delivering and exceeding CTC-like doses of nicotine [12,13]. ECIG nicotine content and delivery profiles are concerning because of nicotine’s ability to promote dependence following experimentation and progression to chronic use [14,15]. ECIGs are also highly customizable, as users may adjust their nicotine levels and device wattage and choose different ECIG liquid flavors [16]. Flavor availability, which is currently unrestricted for ECIGs, may play an important role in CTC cessation patterns and/or maintaining ECIG use patterns among adults [17], as well as potentially attracting youth to begin and continue to use these products [18,19]. Clinical studies have suggested that ECIG flavors increase reward or reinforcing value among current CTC smokers [20,21]. Unfortunately, the range of available ECIG nicotine levels and flavors challenges their individual assessment using controlled clinical studies. 

With the 2016 deeming rule that ECIGs are subject to the regulatory authority of the Center for Tobacco Products of the US Food and Drug Administration (FDA) [1], an opportunity exists to inform the regulation of ECIGs including available nicotine content and flavors to promote the FDA’s “public health standard” [22]. Adhering to this standard requires a consideration of how ECIG regulation, including limiting available ECIG nicotine levels (as in the European Union [23]) or flavors (as with CTCs), will influence the risks and benefits to tobacco users and non-users. Furthermore, per FDA rules, ECIG companies may apply for a designation of “modified risk tobacco product” (MRTP) under either a “modified risk order”, which allows a manufacturer to suggest a product will reduce tobacco-related harms/risks of tobacco-related disease and improve population health, or an “exposure modification order”, which allows a manufacturer to claim a product is free from, has reduced levels of, or reduces exposure to toxins [24]. While no tobacco product has yet achieved an MRTP status, experimental research suggests that claims of reduced risk of harm of a hypothetical MRTP (defined as either an ECIG, heated tobacco product, or snus) elicited lower perceived risk of harm as well as a greater susceptibility of use of the MRTP [25]. In 2017 however, while emphasizing that ECIGs cannot be considered safe, former FDA commissioner Scott Gottlieb announced plans to investigate the role of ECIGs in tobacco harm reduction [26]. Gottlieb extended the agency’s initial deadline scheduled in the agency’s rule of 2016 from August 2018 to August 2022, which will allow ECIG manufacturers more time to submit for FDA approval of their ECIG product [27].

The increased focus of regulatory authorities on the potential role of ECIGs in reducing the public health threat that is CTC smoking, and more importantly the rigorous scientific research that must support this aim, should take the unique characteristics and increasingly modifiable features of ECIGs into consideration, more specifically those that may contribute to their popularity. Thus, scientific evidence predicting how nicotine content, flavor availability, and potential modified risk messaging influence initiation and progression to subsequent ECIG use, as well as perceptions of ECIG harm and addiction, is needed to inform regulatory policies regarding ECIGs. 

Hypothetical choice or preference tasks, where policy-related factors are manipulated, represent one tool for understanding the influence of these ECIG characteristics and messages. Such designs are flexible and can incorporate a variety of measures predictive of tobacco use behaviors including susceptibility to use, attitudes and perceptions, and behavioral economic assessments of abuse liability. For example, an online discrete choice experiment examined the impact of ECIG flavor, device type, and warning label on the likelihood of ECIG use among youth [28]. Findings have suggested that the availability of fruit/sweets/beverage ECIG flavors increased the probability of choosing ECIGs more than other device characteristics, particularly among never users, while warning messages reduced the probability of choosing ECIGs among never users [28]. Another online discrete choice experiment involved three populations differing in tobacco use history—youth and young adult CTC non-smokers, youth and young adult CTC smokers, and adult CTC smokers—and compared preferences for various ECIG characteristics including flavor, nicotine content, health warnings, and price [29]. Here, findings suggested that health warnings produced the strongest effects on ECIG perceptions and intentions for ECIG use [29]. Behavioral economic tasks such as purchase tasks [30] adapted for ECIGs have also been used to examine whether hypothetical changes in ECIG price, concurrent availability of other tobacco products, ECIG flavors, and modified risk messages affect ECIG consumption indices [20,31,32,33]. Generally, results indicated that behavioral economic indices of ECIG use are sensitive to ECIG price manipulations and flavor availability. Taken together, this body of work supports the use of hypothetical policy scenarios to understand the effects of various ECIG characteristics amendable to regulation including nicotine content, flavors, and modified risk messaging on ECIG susceptibility, perceptions, and abuse liability indices—three important predictors of ECIG use behaviors. While various characteristics of ECIGs have been addressed in past studies (e.g.; nicotine, flavors, and health messages), their impact on outcomes indicative of ECIG uptake and subsequent use have often been limited in their scope and have not been compared within a single design among a CTC-smoking and non-smoking population. An important benefit of collecting multiple outcomes within a single design is the ability to provide more robust evidence on hypothetical regulatory environments. 

The goal of this study is to examine how potential regulations on ECIGs will impact a range of outcomes relevant to predicting patterns of ECIG initiation and progression to regular use among adult CTC smokers and non-smokers: susceptibility to use ECIGs, perceptions of ECIG relative harm and addiction, and ECIG abuse liability indices. This study examines several possible regulatory scenarios including regulating the concentration of nicotine in ECIGs, limiting the ECIG flavor options, and allowing modified risk messages on ECIG products indicating reduced harm or reduced exposure to carcinogens compared to CTCs.

## 2. Materials and Methods

### 2.1. Design, Procedures, and Sample

A cross-sectional online experimental survey was conducted among current CTC smokers and non-smokers recruited using Amazon’s Mechanical Turk (MTurk) between January and June 2016. All research procedures were approved by the Virginia Commonwealth University Institutional Review Board as exempt (HM20005750). Participants who were screened and/or completed survey procedures were only identified via their MTurk Worker ID, which was not directly linked to survey data. No other identifiable information was disclosed to or obtained by the researchers. Interested participants responded to an MTurk advertisement and completed a 3-item eligibility survey which assessed age (at least 18 years old), lifetime smoking of 100 CTCs or more (yes, no), and current CTC smoking status (every day, some days, not at all). Eligible current smokers reported smoking at least 100 CTCs throughout their lifetime and reported currently smoking CTCs either every day or occasionally (some days)., while non-smokers reported smoking fewer than 100 CTCs in their lifetime and no current CTC smoking [34]. Former smokers (> 100 CTCs in their lifetime and no current CTC smoking) were ineligible. Regardless of eligibility status, participants were paid $0.05 for filling out the three-item survey. Eligible participants were provided a link to complete an additional survey via Qualtrics. The survey assessed demographics, ever and past 30-day ECIG use (i.e.; “During the past 30 days, which of the following products have you used on at least one day?”, “Electronic Cigarettes or E-cigarettes, such as blu or NJOY”; yes, no), and several other domains prior to randomization to a hypothetical ECIG regulatory condition (described below). At the conclusion of the survey, participants submitted a completion code and received compensation ($2.00) to their MTurk account. 

A total of 1220 individuals completed the eligibility questionnaire, and 1094 completed the survey. Incomplete or suspected duplicate responses (*n* = 66) were removed as well as those with an IP address indicating a non-US location (*n* = 282). The remaining 746 responses were categorized based on their CTC smoking status and past 30-day ECIG use (due to the likely influence of this factor on study outcomes): CTC-only smoker (*n* = 262), dual CTC/ECIG user (*n* = 205), non-CTC/ECIG user (*n* = 272), and ECIG-only user (*n* = 7; excluded from current analyses). These 739 responses were then evaluated for completion of all primary outcomes (*n* = 33 excluded based on abuse liability measures, see detail below) leaving an analytic sample of 706 responses. 

### 2.2. ECIG Regulatory Conditions

After the completion of baseline measures, participants were randomly assigned (with equal probability) to one of eight conditions reflecting three domains of hypothetical ECIG regulatory policy: nicotine content (no nicotine, low nicotine, or high nicotine), flavor (tobacco, fruit, or menthol), and modified risk message (reduced harm or reduced carcinogen exposure relative to CTCs). Condition-specific information was embedded into the questions addressing susceptibility to ECIG use, perceptions of ECIG relative harm and addiction, and ECIG abuse liability indices. No other information/images were presented to participants as part of these conditions (see Appendix A for an example condition).

### 2.3. Measures

#### 2.3.1. Demographics

Participants reported demographic characteristics, including gender (male, female), race/ethnicity (White or Caucasian, Hispanic or Latino, Black or African American, Asian, Middle Eastern, American Indian or Alaskan Native, Native Hawaiian or Other Pacific Islander, Other), age (in years), and education (did not graduate high school, high school graduate, GED, some college or post-high school education, college graduate, some graduate school, graduate degree or higher). Due to the distribution of responses, we recoded participants’ answers for race/ethnicity into three categories (White, Asian, and Other), age into four categories (18–25, 26–29, 30–36, and 37+ years), and education into four categories (high school diploma or less, some college, college graduate, and some graduate school or higher).

#### 2.3.2. Susceptibility to ECIG Use

Following randomization, participants reported susceptibility to using the condition-specific ECIG using an adapted version of the National Cancer Institute Susceptibility to Smoking Questionnaire [35,36]. These four items assessed likelihood of using the condition-specific ECIG soon, in the future, next year, or if offered by a friend. Answers ranged from 1 (definitely yes) to 4 (definitely not). Individuals who responded anything other than “definitely not” to any item were coded as susceptible to condition-specific ECIG use resulting in a binary ECIG susceptibility item (susceptible, not susceptible; original scoring method per [36]). 

#### 2.3.3. Perceptions of ECIG Relative Harm and Addiction

Participants reported perceived relative harm of the condition-specific ECIG condition with the following item: “Compared to regular-strength non-menthol cigarettes, how harmful do you think this product is?” [37] (1 = a lot less harmful, 2 = a little less harmful, 3 = about the same, 4 = a little more harmful, 5 = a lot more harmful, 6=don’t know). The perceived addiction of a condition-specific ECIG was assessed by the following item: “What do you think the likelihood of addiction is when using this product?” [37] (1 = not at all, 2 = slightly, 3 = moderately, 4 = very much, 5 = extremely, 6 = don’t know). Responses of “don’t know” were excluded from analysis.

#### 2.3.4. ECIG Abuse Liability Indices

Finally, participants completed an adapted version of the cigarette purchase task for their condition-specific ECIG to assess abuse liability [20,30]. This task asked participants to provide the number of 10-puff bouts of the condition-specific ECIG they would consume over the course of one day if the 10 puffs cost various prices (16 price points starting at $0.00 and ending at $10.24), given no access to any other tobacco products, and with no ability to save ECIG puffs for future use. We report two outcome measures: the “breakpoint”, the highest price participants would pay for ECIG puffs, and “intensity”, the number of 10-puff bouts participants would consume in a day if the puffs were free. Participants (*n* = 33) were excluded if non-integer consumption values were reported for the purchase task (e.g., a response of ‘0.2 (10-puff bouts in a day)’), if data for any of the 16 prices were missing, or if any increase in consumption at consecutive increasing prices was detected (adapted from the “bounce” criterion for nonsystematic purchase task data) [38].

### 2.4. Data Preparation and Analyses

Following data preparation described above, sample characteristics were then examined by ECIG regulatory domains using bivariate statistics to confirm the equivalence of the sub-samples across domains. We also examined our primary outcomes (susceptibility, perceptions, abuse liability) using descriptive statistics. For ECIG purchase task analyses, breakpoint and intensity outcomes were highly right-skewed. These outcomes were log-transformed after adding a nominal value of 0.01 to retain values of zero.

Regressions (logistic or linear) were fit to the data to test the effects of the ECIG regulatory condition assigned on the outcomes of interest after stratifying by domain (nicotine content, flavor, and message) and CTC/ECIG status. Specifically, the no nicotine ECIG was the referent for the three nicotine content conditions (nicotine domain), the tobacco flavored ECIG was the referent for the three flavor conditions (flavor domain), and the reduced harm ECIG was the referent compared to the reduced carcinogen exposure ECIG in the message conditions (message domain). For susceptibility to condition-specific ECIG use, the sample was restricted to non-CTC/ECIG users and CTC-only smokers due to lack of response variability for dual CTC/ECIG users. For ease of interpretation of the log-outcomes for abuse liability, differences between ECIG regulatory conditions are expressed using percentage differences after exponentiating the regression coefficients using the following formula: (exp(β)−1) × 100. We then tested whether the adjusted associations between ECIG regulatory condition and our outcomes were moderated by CTC/ECIG status by adding an interaction term of CTC/ECIG status and ECIG regulatory condition to each multivariable model using the same analytic strategy described above. All regression models controlled for the demographic characteristics of the participants. All analyses were conducted using STATA 15 (StataCorp, College Station, TX, USA).

## 3. Results

### 3.1. Sample Characteristics

Sample demographics and CTC/ECIG status by ECIG regulatory domain are displayed in Table 1. Bivariate results suggested an equal distribution of gender, age, race/ethnicity, education, and CTC/ECIG use across ECIG regulatory domains (nicotine content, flavor, and message). Overall the sample was slightly more male than female with a majority under 36 years old (68%), White (77%), and reported at least some college education (85%). CTC-ECIG status was split relatively equally across the sample with 38% non-CTC/ECIG users, 35% CTC-only smokers, and 27% dual users. Across all ECIG regulatory domains, 64% of non-users and CTC-only smokers were susceptible to ECIG use (note: dual users were excluded from susceptibility analyses due to a lack of non-susceptible individuals among this sub-group). The mean score for perceived ECIG relative harm was 2.4 (standard deviation 1.0), and the mean score for perceptions of addiction was 3.3 (standard deviation 1.1), each out of a possible five. Before log-transformation of abuse liability outcomes, the median breakpoint was $0.02 (interquartile range 0–1.28), and the median intensity value was 3 (interquartile range 0–10) 10-puff bouts of e-cigarettes.

### 3.2. Susceptibility to ECIG Use

There were no significant differences in susceptibility to ECIG use between the ECIG conditions within the nicotine content domain or the message domain (Table 2). However, for the flavor domain, relative to the tobacco-flavored ECIG condition, the menthol-flavored ECIG condition was associated with significantly lower odds of susceptibility to ECIG use (adjusted odds ratio (AOR = 0.3), *p* < 0.05). Across all three ECIG regulatory domains, CTC smokers had significantly higher odds of ECIG susceptibility than non-CTC/ECIG users (*p*s < 0.001).

### 3.3. Perceived ECIG Relative Harm and Addiction

For the nicotine content domain, relative to the no nicotine ECIG condition, the low nicotine and high nicotine content ECIG conditions were associated with significantly higher ratings of perceived ECIG relative harm (β_low_ = 0.4, β_high_ = 1.2, *ps* < 0.01) and addiction (β_low_ = 1.1, β_high_ = 2.0, *ps* < 0.001; Table 2). For the flavor domain, compared to the tobacco-flavored ECIG condition, the menthol-flavored ECIG condition was associated with significantly higher ratings of perceived ECIG relative harm (β = 0.5, *p* < 0.05). The ECIG message conditions were not significantly associated with perceived ECIG relative harm or addiction. Within the nicotine content and message domains, dual CTC/ECIG user status was associated with significantly lower ratings of perceived ECIG relative harm compared to non-users (*p*s < 0.05); in the flavor domain, CTC-only smoking was associated with significantly greater perceived ECIG relative harm relative to non-users (*p* < 0.05). 

### 3.4. ECIG Abuse Liability Indices

For the nicotine content domain, the low nicotine content ECIG condition was associated with a higher log-breakpoint (β = 0.7, *p* < 0.05) than the no nicotine ECIG condition (Table 2). In other words, the highest price at which participants purchase 10 puffs of a low nicotine content ECIG was about 101% higher (i.e., ((exp(0.7))−1) × 100) than a no nicotine ECIG. For the flavor domain, the menthol-flavored ECIG was associated with a significantly lower log-breakpoint (β = −0.9, *p* < 0.05) and lower log-intensity (β = −1.3, *p* < 0.01) than the tobacco-flavored ECIG. Thus, the maximum price at which participants would purchase a menthol-flavored ECIG was about 59% lower than for a tobacco-flavored ECIG. In regards to intensity, if participants were offered a free menthol-flavored ECIG, they would take 73% fewer 10-puff bouts compared to when offered a free tobacco-flavored ECIG. There were no significant effects of ECIG regulatory condition for the message domain for either log-breakpoint or log-intensity. Across all three domains, CTC-only smokers and dual CTC/ECIG users had significantly greater log-breakpoints and log-intensities compared to non-users (*p*s < 0.001). 

### 3.5. Interactions between ECIG Regulatory Condition and CTC/ECIG Status

Within each domain, we then tested whether the effects of ECIG regulatory conditions on susceptibility, harm and addiction perceptions, and abuse liability differed by CTC/ECIG status (Table 3). For susceptibility to ECIG use (where the sample was restricted to non-users and CTC-only smokers), there were no significant interactions of ECIG regulatory condition and CTC/ECIG status for the nicotine content domain. For the flavor domain, among CTC-only smokers, susceptibility to the fruit-flavored ECIG condition significantly decreased relative to the tobacco-flavored ECIG condition (AOR = 0.1, *p* < 0.01) unlike the effect noted for non-users. For the message domain, among CTC-only smokers, susceptibility to the reduced carcinogen exposure ECIG condition was significantly higher compared to the reduced harm ECIG condition (AOR = 15.3, *p* < 0.05) which differed from non-users. There were no significant interactions of ECIG regulatory condition and CTC/ECIG status for either perceived ECIG relative harm or addiction measures. Among the abuse liability outcomes, for the nicotine content domain, log-breakpoint and log-intensity significantly increased for the high nicotine ECIG condition relative to the no nicotine ECIG condition among CTC-only smokers (log-breakpoint β = 1.7, *p* < 0.05; log-intensity β = 2.3, *p* < 0.05) unlike effects noted for non-users. Regarding the flavor domain, abuse liability estimates for the menthol-flavored ECIG condition were significantly lower than in the tobacco-flavored ECIG condition for CTC-only smokers (log-breakpoint β = −1.7, *p <* 0.05; log-intensity β = −2.2, *p* < 0.05) but not for non-users. Finally, the fruit-flavored ECIG condition influenced log-intensity among CTC-only smokers with significantly lower estimates relative to the tobacco-flavored ECIG condition (β = −2.7, *p* < 0.05) which differed from non-users. Of note, no significant interactions were identified for dual CTC/ECIG users for ECIG relative harm/addiction perceptions or abuse liability indices.

## 4. Discussion

Results from this online randomized design suggest that three ECIG regulatory domains—nicotine content, flavors, and modified risk messages—can influence measures of susceptibility to ECIG use, perceived ECIG relative harm and addiction, and ECIG abuse liability indices, and many of these effects were moderated by CTC/ECIG status. ECIG nicotine content and flavor conditions were most influential with significant main effects of ECIG regulatory condition and/or interactions with CTC/ECIG status noted for almost every outcome assessed. Overall greater nicotine content and tobacco flavor ECIGs were more highly preferred among CTC smokers relative to non-users. These findings reinforce other reports [28,29] regarding the importance of ECIG characteristics in patterns of ECIG uptake and subsequent use among CTC smoking and non-smoking populations. The findings also highlight novel differences in the effects of hypothetical ECIG regulatory conditions between groups differing in tobacco use history. 

Within the nicotine content regulatory domain, main effects of condition were observed for most outcomes and interactions with CTC/ECIG status were observed for abuse liability indices. Relative to an ECIG containing no nicotine, an ECIG containing higher levels of nicotine was associated with greater perceived ECIG relative harm, perceived ECIG addiction, and log-breakpoint for ECIGs. The interactions for log-breakpoint and log-intensity and CTC/ECIG status suggested that the high nicotine content ECIG condition appealed more to CTC-only smokers than non-users. These results suggest awareness of nicotine as a potentially harmful constituent of ECIGs consistent with previous research regarding CTC ingredients [39]. Previous studies have suggested that smokers falsely identify nicotine as the facilitator of detrimental health effects [40,41]. Additionally, other research suggests that those who have never smoked were more likely to perceive nicotine as the cause of cancers due to smoking when compared to current smokers and quitters [42]. Despite this greater perceived relative harm/addiction, nicotine-containing conditions remained more appealing among CTC-only smokers which may be due to the dependence characteristics among this group. 

Within the flavor domain, relative to a tobacco-flavored ECIG, menthol was associated with lower susceptibility for ECIG use, greater perceived ECIG relative harm, and lower abuse liability. Menthol-related findings may reflect the general perception of harm regarding menthol-flavored tobacco products evidenced by greater perceived risk of menthol CTCs relative to nonmenthol CTCs among other studies [43]. Interactions between ECIG regulatory condition and CTC-ECIG status revealed lower preferences for the menthol and fruit-flavored ECIG conditions relative to the tobacco-flavored ECIG condition but only among CTC-only smokers. This finding aligns with prior evidence among CTC-only smokers where tobacco-flavored ECIGs were more highly preferred to other flavors in a discrete choice experiment [29] as well as a clinical laboratory study where tobacco-flavored ECIGs were more similar to own-brand CTCs on purchase task measures of abuse liability [20]. The lack of interest among CTC-only smokers for non-tobacco-flavored ECIGs may reflect preferences for ECIG products that more closely simulate CTC use [44]. Results on menthol and fruit flavors among this sample support regulation on ECIG flavors to prevent initiation by vulnerable populations, such as youth and non-smokers, while preserving the potential for CTC-only smokers to transition to ECIGs. 

Finally, within the message domain, there were no significant main effects of ECIG regulatory condition (reduced ECIG harm vs. reduced ECIG carcinogen exposure) across outcomes suggesting participants interpreted these messages similarly. However, the interaction results revealed CTC-only smokers were more susceptible to the reduced carcinogen exposure ECIG relative to non-users. While these condition-specific effects were sparse, findings from this study are germane to FDA’s decision in 2018 to include nicotine warning labels for tobacco products covered by the Tobacco Control Act, including nicotine-containing ECIG liquids. The labels are required to state only: “WARNING: This product contains nicotine. Nicotine is an addictive chemical” [45]. Simply advising consumers that nicotine is addictive may not be sufficient to change tobacco product uptake and use. For example, our results suggest that nicotine content labeling should take into account differences between CTC-only smokers and non-users in perceptions of and susceptibility to nicotine-containing ECIGs, and education efforts should provide clear messaging about the relative harms and different methods used to deliver nicotine [46]. Future work should also include more specific harm outcomes/exposure information as specificity is linked with effectiveness [47], as well as qualitative measures to fully characterize individual responses to potential modified risk messaging.

### Limitations

The current sample has several characteristics that limit generalizability including greater use of technology and internet access, familiarity with MTurk, and demographic distribution. Our data collection took place online; thus, all of our participants had access to an internet-enabled device and were sufficiently familiar with maneuvering such devices as well as the internet. Moreover, self-selection likely factored into the composition of the final sample, as participants may have participated due to their interest in the task at hand, the topic, and/or the monetary incentive offered [48]. Out of the total sample, 8.3% described themselves as Asian, whereas across the US population, only 5.6% identify as Asian [49]. This difference is reinforced by others that suggest MTurk may provide more diverse samples compared to US college populations or other online sources [50]. Over half of our participants had graduated from college, which suggests a highly educated sample that may not be representative of the general US population, out of which only 33% hold a bachelor’s degree or a higher degree [51]. Due to these limitations and space considerations, the results of adjusted associations for demographic covariates were not included in Table 2 but are available in Appendix A. An examination of interactions between demographics and preferences for ECIG characteristics may be an important area for future work in this area. Importantly, crowdsourcing services have been reliably utilized in past research [52] including tobacco research [53,54,55]. Susceptibility and perception responses also may have been influenced by respondents’ previous experience with ECIGs and the majority of current smokers (60.2%) reported ever use of ECIGs relative to 11.2% of non-smokers. However, our categorization of the sample by past 30-day ECIG status allowed us to examine the effects of potential ECIG regulatory conditions inclusive of this factor. In order to allow for further generalizability, future studies may consider constructing more representative samples of tobacco users and non-users, including former smokers.

## 5. Conclusions

Increasing uptake of ECIGs and product diversity in the ECIG market have challenged regulatory efforts to minimize the public health burden of tobacco products. Our findings suggest that modifications to regulations regarding nicotine levels, flavor availability, and harm messaging may impact the susceptibility of ECIG use, harm and addiction perceptions, and abuse liability to varying degrees. Furthermore, CTC-only smokers and non-users differ in their perceptions, experiences, and motivations associated with tobacco product use; thus, regulations on ECIG characteristics may affect these groups in different ways. For example, differences in perceptions across nicotine levels may reflect misperceptions of nicotine as a harmful constituent, which may influence the risk of ECIG initiation and continued use among different tobacco product user groups. Additionally, differences in fruit-flavored ECIG susceptibility and abuse liability indices between CTC-only smokers and non-users indicate that regulations to restrict ECIG flavors may prevent initiation among non-users—potentially including youth—without impacting initiation among CTC-only smokers. Education regarding the absolute and relative harms of ECIGs, including differences by nicotine content and flavors, will be a vital complement to future regulations. Looking ahead, realizing the full potential of tobacco regulatory actions to improve public health will depend in part upon the extent to which ECIG policies consider heterogeneous influences across tobacco product user groups.

## Figures and Tables

**Table 1 ijerph-16-01825-t001:** Sample Characteristics by ECIG Regulatory Domain.

	Overall *n* = 706		Nicotine Content *n* = 256		Flavor *n* = 264		Message *n* = 186		
Variables	*n*	%	*n*	%	*n*	%	*n*	%	*p*
**ECIG Regulatory Condition**									N/A
No nicotine ECIG	90	12.8	90	35.2	-	-	-	-	
Low nicotine ECIG	88	12.5	88	34.4	-	-	-	-	
High nicotine ECIG	78	11.1	78	30.5	-	-	-	-	
Tobacco-flavored ECIG	81	11.5	-	-	81	30.7	-	-	
Menthol-flavored ECIG	86	12.2	-	-	86	32.6	-	-	
Fruit-flavored ECIG	97	13.7	-	-	97	36.7	-	-	
Reduced harm ECIG	91	12.9	-	-	-	-	91	48.9	
Reduced CE ECIG	95	13.5	-	-	-	-	95	51.1	
**Gender**									0.154
Male	383	54.3	149	58.2	143	54.2	91	48.9	
Female	323	45.8	107	41.8	121	45.8	95	51.1	
**Age (years)**									0.833
18–25	134	19.0	46	18.0	51	19.3	37	19.9	
26–29	169	23.9	55	21.5	67	25.4	47	25.3	
30–36	179	25.4	67	26.2	63	23.9	49	26.3	
37+	224	31.7	88	34.4	83	31.4	53	28.5	
**Race/ethnicity**									0.519
White/Caucasian	540	77.1	197	77.6	204	78.2	139	75.1	
Asian	58	8.3	19	7.5	18	6.9	21	11.4	
Other	102	14.6	38	15.0	39	14.9	25	13.5	
**Education**									0.157
High school/GED or below	107	15.2	51	19.9	34	12.9	22	11.9	
Some college	234	33.2	79	30.9	88	33.3	67	36.2	
College graduate	275	39.0	99	38.7	102	38.6	74	40.0	
Post-college education	89	12.6	27	10.6	40	15.2	22	11.9	
**CTC/ECIG Status**									0.756
Non-CTC/ECIG smoker	267	37.8	95	37.1	101	38.3	71	38.2	
CTC-only smoker	246	34.8	92	35.9	85	32.2	69	37.1	
Dual CTC/ECIG user	193	27.3	69	27.0	78	29.6	46	24.7	

Note: CE = Carcinogen Exposure; CTC = Combustible Tobacco Cigarette; ECIG = Electronic Cigarette. GED = General Education Diploma. Items with missing cases included race/ethnicity (nicotine content *n* = 2, flavor domain *n* = 3, and message domain *n* = 1) and education (message domain *n* = 1).

**Table 2 ijerph-16-01825-t002:** Adjusted Associations among the Electronic Cigarette (ECIG) Regulatory Conditions and Susceptibility, Perceptions, and Abuse Liability Indices.

	Susceptibility to ECIG Use		Perceived ECIG Relative Harm		Perceived ECIG Addiction		Log-Breakpoint		Log-Intensity	
	AOR (95% CI)	*p*	β (95% CI)	*p*	β (95% CI)	*p*	β (95% CI)	*p*	β (95% CI)	*p*
**Nicotine Content Domain**	*n* = 186		*n* = 245		*n* = 245		*n* = 254		*n* = 254	
ECIG Regulatory Condition										
No nicotine ECIG	Ref		Ref		Ref		Ref		Ref	
Low nicotine ECIG	2.2 (0.9, 5.7)	0.098	**0.4 (0.1, 0.7)**	**0.006**	**1.1 (0.9, 1.4)**	**<0.001**	**0.7 (0.1, 1.3)**	**0.034**	0.7 (−0.2, 1.6)	0.107
High nicotine ECIG	1.6 (0.6, 4.2)	0.341	**1.2 (0.9, 1.5)**	**<0.001**	**2.0 (1.7, 2.2)**	**<0.001**	0.5 (−0.2, 1.2)	0.143	0.3 (−0.6, 1.2)	0.479
CTC/ECIG Status										
Non-CTC/ECIG user	Ref		Ref		Ref		Ref		Ref	
CTC-only smoker	**29.8 (11.3, 78.1)**	**<0.001**	−0.3 (−0.6, 0.0)	0.097	−0.1 (−0.4, 0.2)	0.674	**2.9 (2.3, 3.6)**	**<0.001**	**4.1 (3.2, 5.1)**	**<0.001**
Dual CTC/ECIG user	-^a^	-^a^	**−0.5 (−0.8, −0.1)**	**0.005**	−0.3 (−0.6, 0.0)	0.087	**3.6 (2.9, 4.3)**	**<0.001**	**5.6 (4.6, 6.6)**	**<0.001**
**Flavor Domain**	*n* = 184		*n* = 247		*n* = 247		*n* = 261		*n* = 261	
ECIG Regulatory Condition										
Tobacco-flavored ECIG	Ref		Ref		Ref		Ref		Ref	
Menthol-flavored ECIG	**0.3 (0.1, 0.8)**	**0.020**	**0.5 (0.2, 0.8)**	**0.001**	0.3 (−0.1, 0.6)	0.136	**−0.9 (−1.6, −0.3)**	**0.008**	**−1.3 (−2.2, −0.5)**	**0.002**
Fruit-flavored ECIG	0.8 (0.3, 1.8)	0.548	0.1 (−0.2, 0.4)	0.603	−0.1 (−0.4, 0.2)	0.597	−0.1 (−0.8, 0.5)	0.663	0.2 (−0.6, 1.0)	0.612
CTC/ECIG Status										
Non-CTC/ECIG user	Ref		Ref		Ref		Ref		Ref	
CTC-only smoker	**9.6 (4.5, 20.4)**	**<0.001**	**0.4 (0.1, 0.7)**	**0.004**	0.1 (−0.2, 0.4)	0.546	**2.3 (1.7, 3.0)**	**<0.001**	**3.8 (2.9, 4.6)**	**<0.001**
Dual CTC/ECIG user	-^a^	-^a^	−0.2 (−0.4, 0.1)	0.259	0.0 (−0.3, 0.3)	0.925	**3.4 (2.7, 4.1)**	**<0.001**	**5.5 (4.6, 6.3)**	**<0.001**
**Message Domain**	*n* = 138		*n* = 177		*n* = 174		*n* = 184		*n* = 184	
ECIG Regulatory Condition										
Reduced harm ECIG	Ref		Ref		Ref		Ref		Ref	
Reduced CE ECIG	0.4 (0.1, 1.1)	0.075	0.2 (−0.1, 0.4)	0.173	0.2 (−0.1, 0.4)	0.232	0.2 (−0.4, 0.8)	0.542	−0.4 (−1.1, 0.3)	0.255
CTC/ECIG Status										
Non-CTC/ECIG user	Ref		Ref		Ref		Ref		Ref	
CTC-only smoker	**43.9 (11.9, 161.6)**	**<0.001**	0.2 (−0.1, 0.4)	0.253	−0.2 (−0.5, 0.1)	0.170	**3.7 (3.0, 4.4)**	**<0.001**	**5.0 (4.1, 5.8)**	**<0.001**
Dual CTC/ECIG user	-^a^	-^a^	**−0.6 (−0.9, −0.3)**	**<0.001**	−0.2 (−0.5, 0.2)	0.284	**4.2 (3.4, 5.0)**	**<0.001**	**6.9 (6.0, 7.8)**	**<0.001**

Note: All regression models control for participant gender, race/ethnicity, age, and education. CE = carcinogen exposure; CTC = Combustible Tobacco Cigarette; ECIG = e-cigarette; AOR = Adjusted Odds Ratio; **Bold** values indicate statistical significance (*p* < 0.05). ^a^ Due to cell size frequency, this response category was not included in analyses for the susceptibility to ECIG use outcome.

**Table 3 ijerph-16-01825-t003:** Adjusted Interaction Results of Electronic Cigarette (ECIG) Regulatory Condition and Tobacco User Status for Susceptibility and Abuse Liability Indices.

	Susceptibility to ECIG Use		Log-Breakpoint		Log-Intensity	
	AOR (95% CI)	*p*	β (95% CI)	*p*	β (95% CI)	*p*
**Nicotine Content Domain**	*n* = 186		*n* = 254		*n* = 254	
ECIG Regulatory Condition * CTC/ECIG Status						
No nicotine ECIG * Non-CTC/ECIG user	Ref		Ref		Ref	
Low nicotine ECIG * CTC-only smoker	2.3 (0.3, 15.6)	0.399	1.2 (−0.3, 2.7)	0.107	1.8 (−0.3, 3.8)	0.089
Low nicotine ECIG * Dual CTC/ECIG user	-^a^	-^a^	1.3 (−0.4, 3.0)	0.129	−1.1 (−3.4, 1.3)	0.367
High nicotine ECIG * CTC-only smoker	4.2 (0.5, 34.4)	0.183	**1.7 (0.2, 3.2)**	**0.031**	**2.3 (0.2, 4.4)**	**0.029**
High nicotine ECIG * Dual CTC/ECIG user	-^a^	-^a^	1.0 (−0.7, 2.7)	0.243	−0.4 (−2.7, 1.9)	0.750
**Flavor Domain**	*n* = 184		*n* = 261		*n* = 261	
ECIG Regulatory Condition * CTC/ECIG Status						
Tobacco-flavored ECIG * Non-CTC/ECIG user	Ref		Ref		Ref	
Menthol-flavored ECIG * CTC-only smoker	0.2 (0.0, 1.5)	0.106	**−1.7 (−3.3, −0.1)**	**0.041**	**−2.2 (−4.2, −0.2)**	**0.034**
Menthol-flavored ECIG * Dual CTC/ECIG user	-^a^	-^a^	−0.8 (−2.5, 0.8)	0.316	−0.2 (−2.3, 1.9)	0.851
Fruit-flavored ECIG * CTC-only smoker	**0.1 (0.0, 0.4)**	**0.004**	−1.5 (−3.0, 0.1)	0.063	**−2.7 (−4.6, −0.8)**	**0.006**
Fruit-flavored ECIG * Dual CTC/ECIG user	-^a^	-^a^	−0.0 (−1.7, 1.6)	0.968	−0.2 (−2.3, 1.9)	0.866
**Message Domain**	*n* = 138		*n* = 184		*n* = 184	
ECIG Regulatory Condition * CTC/ECIG Status						
Reduced harm ECIG * Non-CTC/ECIG user	Ref		Ref		Ref	
Reduced CE * CTC-only smoker	**15.3 (1.1, 212.1)**	**0.042**	1.4 (0.0, 2.7)	0.053	1.6 (−0.0, 3.2)	0.057
Reduced CE * Dual CTC/ECIG user	-^a^	-^a^	0.6 (−0.9, 2.2)	0.411	0.6 (−1.1, 2.4)	0.466

Note: All regression models control for participant gender, race/ethnicity, age, and education and include main effects of ECIG regulatory condition and CTC/ECIG status. No significant interactions were noted for perceived ECIG relative harm or addiction and thus are not included here. CE = carcinogen exposure; CTC = Combustible Tobacco Cigarette; ECIG = e-cigarette; AOR = Adjusted Odds Ratio; **Bold** values indicate statistical significance (*p* < 0.05). ^a^ Due to cell size frequency, this response category was not included in analyses for the susceptibility to ECIG use outcome.

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
