# Peer review of "Influence of Electronic Cigarette Characteristics on Susceptibility, Perceptions, and Abuse Liability Indices among Combustible Tobacco Cigarette Smokers and Non-Smokers"

_ijerph, 2019, doi:10.3390/ijerph16101825_

Round 1

Reviewer 1 Report

The manuscript, “Influence of Electronic Cigarette Characteristics on Susceptibility, Perceptions, and Abuse Liability Indices among Combustible Tobacco Cigarette Smokers and Non-Smokers” is a very well written and thoroughly designed study examining e-cigarette susceptibility, harm/addiction, and abuse liability as a function of hypothetical regulatory domains (nicotine content, flavor, message). Each outcome was affected by the regulatory domain scenarios in various ways and most interacted with smoker status.

1. Did you find any interaction between your outcome variables and the demographics of the participants? For example, was there any age difference in the e-cig users who used flavored liquid compared to those who used tobacco or menthol flavor (e.g., flavored users are younger)? If so, does age differentially affect susceptibility, harm/addiction, and/or abuse liability?

2. As a minor point on style – the results and discussion sections are very dense (packed with good information) though leaving them sometimes hard to follow. Depending on length restrictions, adding in a few extra explanatory sentences and/or phases throughout to improve readability would be preferred.

Author Response

The authors wish to thank reviewers and the editor for their helpful comments to improve the contribution of this manuscript.  We have made every effort to address each comment.  Below each reviewer comment, our response is included in italics. We look forward to hearing any feedback regarding this revision.

Reviewer 1

1. Did you find any interaction between your outcome variables and the demographics of the participants? For example, was there any age difference in the e-cig users who used flavored liquid compared to those who used tobacco or menthol flavor (e.g., flavored users are younger)? If so, does age differentially affect susceptibility, harm/addiction, and/or abuse liability?

We thank the reviewer for this comment. As described in Table 2 each of our regression models controlled for gender, race/ethnicity, age, and education. Resulting estimates from these covariates are not included in the current manuscript for space concerns as well as consideration of the population sampled (not representative of US).We have now included the full results of all covariates tested in the supplemental material (S1-S3) and referenced these tables in the discussion. Broadly they suggest demographic associations were sparse and not very consistent. Interactions between e-cigarette regulatory domains and participant demographics were not tested in these regressions due to the sample sizes available for most demographic sub-categories (see Table 1), but we have added this as a suggestion for future research in this area.

2. As a minor point on style – the results and discussion sections are very dense (packed with good information) though leaving them sometimes hard to follow. Depending on length restrictions, adding in a few extra explanatory sentences and/or phases throughout to improve readability would be preferred.

We believe several changes made in response to other reviewer comments as well as an additional edit to the discussion section have addressed this reviewer concern. We hope these changes make the document more readable throughout.

Reviewer 2 Report

General comments

This is an original cross-sectional study which aims to explore how three different regulatory scenarios (in relation to nicotine content, flavor, and harm messaging) of the electronic cigarettes (ECIG) may impact on three different outcomes of initiation or regular use of ECIG (related to susceptibility, harm/addiction perceptions, and abuse liability) among current combustible tobacco cigarette (CTC) smokers and nonsmokers.

Main conclusion is that regulations about nicotine content, flavor, and harm messaging may impact on susceptibility of ECIG use, harm and addiction perceptions, and abuse liability to varying degrees.

A relatively small sample (n=706), in relation to the aim of the study, is analyzed. Methodological concerns need to be addressed. The topic of the research is of interest and appropriate for the Special Issue "Electronic Cigarettes: Good and Bad Impacts" of the Journal. Overall, the paper is not easy to follow and too long, taking into account that it is an original paper.

Specific comments

Major

- There is discrepancy in the reported number of the study subjects. In the abstract and in table 1 it is reported that n=706 subjects completed the online survey. In the Materials and Methods section, it is reported that finally n=739 were categorized for subsequent analyses. However, when n=66 subjects for incomplete or suspected duplicate responses and n=282 subjects for an IP address indicating a non-US location are removed from the n=1094 subjects who completed the survey, a total sample of n=746 subjects is obtained. (See page 3, lines 119-121).

- Due to the online enrollment by using the Amazon’s Mechanical Turk, a self-selection bias cannot be excluded. The authors partly discuss this limit in the discussion. It should be taken into account also the confounder of the income of the study subjects; indeed, the subjects who completed the survey received a money compensation.

- While regulatory authorities should be provided with scientific evidences for modifying regulation of ECIGs, the results from the present study seem to be intuitive and general. Thus, a more clear and specific message should be provided in the conclusions section of the Abstract and text. 

- The paper lacks of recent references on available data regarding safety and public health impacts of ECIG and the updated status of US regulations and policies for their sale and use.

- In the Abstract, main results should be reported also as numeric data.

Minor

- Abstract. page 1, line 16. Add "combustible tobacco" before "cigarette".

Author Response

The authors wish to thank reviewers and the editor for their helpful comments to improve the contribution of this manuscript.  We have made every effort to address each comment.  Below each reviewer comment, our response is included in italics. We look forward to hearing any feedback regarding this revision.

Reviewer 2

Major

3. There is discrepancy in the reported number of the study subjects. In the abstract and in table 1 it is reported that n=706 subjects completed the online survey. In the Materials and Methods section, it is reported that finally n=739 were categorized for subsequent analyses. However, when n=66 subjects for incomplete or suspected duplicate responses and n=282 subjects for an IP address indicating a non-US location are removed from the n=1094 subjects who completed the survey, a total sample of n=746 subjects is obtained. (See page 3, lines 119-121).

Thank you for making us aware of the unclear presentation of how we derived our final sample size. We hope to have fixed this issue in the abstract and the Design, Procedures, and Sample section by organizing information related to participant exclusion in a more logical manner so the reviewers and readers can follow our steps.

4. Due to the online enrollment by using the Amazon’s Mechanical Turk, a self-selection bias cannot be excluded. The authors partly discuss this limit in the discussion. It should be taken into account also the confounder of the income of the study subjects; indeed, the subjects who completed the survey received a money compensation.

We thank the reviewer for this comment. We have added an additional sentence in the limitations section regarding this potential self-selection bias and a relevant reference regarding this phenomena among MTurk respondents.

5.  While regulatory authorities should be provided with scientific evidences for modifying regulation of ECIGs, the results from the present study seem to be intuitive and general. Thus, a more clear and specific message should be provided in the conclusions section of the Abstract and text. 

We have made specific changes to the Abstract and Discussion sections to reinforce the primary findings of this study.

6. The paper lacks of recent references on available data regarding safety and public health impacts of ECIG and the updated status of US regulations and policies for their sale and use.

We have made significant changes to the introduction section to address this concern.

7. In the Abstract, main results should be reported also as numeric data.

Due to the word count limitations for the abstract (200) and large number of numeric results for this manuscript we were unable to make this change but we believe the results summary in the abstract provides an accurate and succinct description of the findings observed.

Minor

8. Abstract. page 1, line 16. Add "combustible tobacco" before "cigarette".

The requested change has been made.

Round 2

Reviewer 2 Report

The authors have adequately answered to the reviewers' concerns and modified the paper accordingly. This reviewer has no more comments.